# Oral Toxicity and Hypotensive Influence of Sericin-Derived Oligopeptides (SDOs) from Yellow Silk Cocoons of *Bombyx mori* in Rodent Studies

**DOI:** 10.3390/foods13213505

**Published:** 2024-11-01

**Authors:** Chainarong Tocharus, Virakboth Prum, Manote Sutheerawattananonda

**Affiliations:** 1Department of Anatomy, Faculty of Medicine, Chiang Mai University, Chiang Mai 50200, Thailand; chainarongt@hotmail.com; 2School of Food Technology, Institute of Agricultural Technology, Suranaree University of Technology, Nakhon Ratchasima 30000, Thailand; prumvirakboth9@gmail.com

**Keywords:** sericin hydrolysate, oral toxicity, hypotensive effect, silk protein, yellow silk cocoons

## Abstract

Sericin-derived oligopeptides (SDOs) from yellow silk cocoons exhibit antihypertensive and hypoglycemic properties in both in vitro and in vivo studies. This study investigated the acute toxicity of SDOs as a novel food for human consumption using female ICR mice and Wistar rats, as well as the chronic toxicity test on both sexes of Wistar rats. Clinical chemistry, hematology, and histopathological studies revealed that SDOs were safe for a single dose of 2000 $\mathrm{mg}\;kg^{-1}$ body weight (BW) and daily oral administration of 50, 100, and 200 $\mathrm{mg}\;kg^{-1}$ BW for six months. The chronic toxicity study additionally measured the rats’ systolic blood pressure (SBP) and blood sugar monthly as they slowly aged. In the 2nd month for male rats and the 4th month for both sexes, SDOs had a significant hypotensive effect on Wistar rats’ blood pressure, lowering it from 130 mmHg to a plateau at 110–115 mmHg. In contrast, the blood pressure of the control rats exceeded 140 mmHg after five months. Nonetheless, the hypoglycemic effect was not observed. Measurements of SBP and blood glucose in aged rats during chronic toxicity tests yielded insights beyond ordinary toxicity, including the health and fitness of the lab rats, perhaps resulting in novel discoveries or areas of study that justify the sacrifice of the animals’ lives.

## 1. Introduction

Sericin is one of the two most important proteins produced by silkworms to form silk cocoons. It functions as an adhesive to bind two strains of fibroin together and a protective layer for silk threads from the adverse environment [1,2]. Apart from the white silkworms, yellow silkworms, typically raised in tropical climates, have specific DNA that can accumulate lutein from mulberry leaves in their silk glands by binding with sericin, resulting in golden-yellow silk cocoons [3,4]. This may, in part, contribute to sericin’s efficacy and biological activities. Researchers discovered that sericin, sericin hydrolysates, and SDOs from yellow silk cocoons had hypoglycemic, antihypertensive, antioxidant, cholesterol-lowering, and colon cancer-reducing properties [5,6,7,8,9,10,11,12]. Sericin is usually discarded along with wastewater during the silk reeling process. In the textile industry, yellow silk cocoons, with their short silk threads and low yield of raw silk yarns from the high composition of sericin (up to 35% dry weight basis), are considered inferior to white cocoons [13]. This inferiority, however, may be exploited to produce protein alternatives such as silk sericin, sericin-derived products, and protein-binding lutein from yellow silk cocoons [14,15,16]. Food ingredients from the sericulture of white silk cocoons, such as fibroin hydrolysate, silkworm powder, and acid-hydrolyzed silk cocoon powder, have been researched and are available on the market [17,18,19,20]. One cycle of silk production typically takes one month, from egg hatching to silk yarn reeling [21]. Silkworms feed exclusively on organic mulberry leaves, providing an alternative source of organic insect protein that is considered organic, environmentally friendly, sustainable, self-sufficient, and valuable if officially approved for human consumption [18,19,22,23,24,25,26].

SDOs have been shown to be more effective for biological activities than intact sericin. SDOs were found to inhibit the activity of ACE and/or DPP-IV in an in vitro model [7,27]. Their activity was maintained after simulated GI digestion, but after digestion with plasmin, a blood protease, the anti-ACE activity increased significantly [7]. However, there was a slight decrease in anti-activity for DPP-IV. In vitro studies have shown that sericin hydrolysates inhibited pancreatic α-amylase and α-glucosidase activity in the jejunum and duodenum of the small intestine [28,29]. The smaller molecular weight of sericin hydrolysate may help resist enzymatic digestion in the GI tract, partially absorb it into the blood circulation system [13,28], and facilitate its efficient activity as it passes through the first part of the small intestine. The abovementioned studies often focus on the in vitro ACE inhibition activity of SDOs or sericin hydrolysates; however, the vasorelaxant activity, a different mechanism associated with the blood pressure-lowering effect of SDOs, has also been investigated. In an ex vivo study, SDOs from yellow silk cocoons demonstrated a blood pressure-lowering ability via NO/sGC/cGMP signal pathways in the endothelium and a reduction in calcium influx in smooth muscle [6]. The hypoglycemic property of SDOs in STZ-induced rats was discovered possibly through the pathways of anti-activity of pancreatic enzymes: α-amylase and α-glucosidase and the competition for DPP-IV present in the blood system with glucagon-like peptide-1 (GLP-1) [5]. This could help the beta cell regenerate and become active in insulin production again, resulting in decreased blood sugar [30,31]. Existing data show that sericin and sericin hydrolysate, including SDOs obtained from yellow silk cocoons, are effective, but there are not enough data suggesting that they are safe for human consumption.

Although the sericulture community in Thailand’s northeastern region has traditionally consumed silk pupae as a protein source, and they are now available as street food in several tourist destinations throughout Thailand, sericin has not been widely consumed by locals [32,33]. Given that sericin, particularly SDOs derived from yellow silk cocoons, has been shown to have numerous health benefits and efficacy in relation to metabolic syndromes, regular consumption by a healthy individual may either improve or degrade overall health. It is considered a novel component of the human diet. Safety data obtained through animal testing must be provided to comply with the stringent food safety guidelines for a novel food. In this study, the safety of SDO consumption was examined following the Organization for Economic Co-operation and Development (OECD) guidelines no. 420 in preference to other methods [34,35,36,37,38].

For acute oral toxicity, the EEC hazard classification system uses three toxic categories (as measured in $\mathrm{mg}\;kg^{-1}$): less than 25, >25–200, and greater than 200–2000. We employed the fixed-dose procedure (FDP), which the OECD approved in 1992 by incorporating a sighting study into the original FDP protocol, to minimize the number of test animals and prevent lethality as an endpoint. The FDP uses clear toxic signs at fixed-dose levels that correspond to the EEC acute toxicity classification system, rather than relying on lethality. Based on the sighting study for FDP (OECD Test Guideline 420), classification decisions can be made for various reasons such as evident toxicity, moribundity, and mortality [39]. The OECD-recommended modification made the test results applicable to all classification systems. The FDP tests typically use either 10 or 20 animals in total, while conventional LD50 determination typically requires 30 animals (15 for each sex) [40].

Although the FDP is administered to both sexes, the responses in acute oral toxicity tests are typically similar, with females tending to exhibit a higher level of sensitivity than males [41,42]. Previous studies have conducted acute oral toxicity tests on females of one or two different species for various plant extracts [43,44,45,46], except for the study by Li et al. (2020) [41], which used both sexes of two species.

Given that SDOs from yellow silk cocoons are considered a novel substance, little to no existing toxicological studies have been conducted on them; therefore, the use of two species can be more comprehensive for safety evaluation, reducing the uncertainty associated with novel compounds. This would help in identifying any unexpected toxic effects that might only be detectable in one species. In addition, according to the Food and Drug Administration (FDA), more rigorous testing for novel substances based on testing results from two species is often required to ensure their safety, especially when there is a lack of prior human consumption data [47]. This approach would help in building a strong case for the safe introduction of SDOs into the food supply for the general public and understanding the mode of action, which can be crucial in predicting the long-term safety and potential health benefits.

This study examined acute toxicity in female ICR mice and Wistar rats, alongside chronic toxicity in both sexes of Wistar rats. Additionally, the impact of prolonged SDO consumption on the rats’ blood glucose and blood pressure levels was assessed throughout the chronic toxicity testing period to evaluate the cumulative effects of SDOs as the rats aged gradually.

## 2. Materials and Methods

### 2.1. Sericin-Derived Oligopeptide (SDO) Preparation

The protocol developed by Sutheerawattananonda et al. (2013) [48] was used to prepare SDOs. Sericin was dissolved by autoclaving yellow silk cocoons of the *Bombyx mori* Nang Noi Srisaket-1 variety, which were obtained from Queen Sirikit Sericulture Centers in Chaiyaphum, Thailand, for a duration of 30 min. The extracted cocoons were separated from the liquid by filtering the sericin-rich protein solution using cheesecloth. The sericin solution was subjected to enzymatic hydrolysis using protease (16 U/g, EC no. 2327522, Sigma, St. Louis, MO, USA). A 1 mL solution of protease enzyme, containing 0.01 unit/mL of protease enzyme in a 0.036 M calcium chloride ($CaCl_{2}$) solution at a 1:1 volumetric ratio, was mixed with 300 mL of the obtained sericin solution and agitated for 1 h at a temperature of 37 °C. The solution was subsequently heated to a temperature of 90 °C for a period of 15 min to deactivate the enzymatic activity. It was then cooled to room temperature and subjected to centrifugation at a force of 9500× *g* for 15 min at a temperature of 4 °C to separate the solids. The process involved the utilization of a hollow fiber membrane with a molecular weight cut-off (MWCO) of 5000, allowing for the separation of oligopeptides with molecular weights below 5 kDa from larger oligopeptides through ultra-membrane filtration using equipment from GE Healthcare Bio-Sciences AB in Uppsala, Sweden.

The resulting SDO solution was subjected to freeze-drying and stored in an airtight container at room temperature until it was used. SDOs were sent to Central Laboratory (Thailand) Co., Ltd., Khon Kaen, Thailand for analysis of trace metals using the in-house method based on EPA 3052. Microbiological contamination includes Coliforms and *E. coli* (FDA BAM Online, 2017), *Listeria monocytogenes* (ISO 11290-1:2017 (E)) [49], *Salmonella* spp. (ISO 6579-1:2017 (E)) [50], *Staphylococcus aureus* (FDA BAM Online, 2016 (Chapter 12)) [51], total plate count (FDA BAM Online, 2001 (Chapter 3)) [52], and Yeasts and Molds (FDA BAM Online 2001, Chapter 18) [53]. Pesticide residues were analyzed using an in-house method based on QuEChERS (Method EN 15662:2018) [54], carbamate groups (LC-MS), organochlorine and organophosphate groups (GC/ECD), and pyrethroid groups (GC/FPD). Table 1 shows SDO specifications for yellow silk cocoons used in this study. We made SDO solutions daily by mixing SDO powder with distilled water to provide the required SDO concentrations for oral administration (oral gavage) to the animals.

### 2.2. Animals

For this study, the National Bureau of Laboratory Animals at Mahidol University in Salaya, Nakhon Pathom, Thailand, provided 24 male and 36 female Wistar rats (12 for acute toxicity test and 24 for chronic toxicity test) weighing 180–200 g and 12 female ICR mice weighing 20–30 g at 8 weeks of age. They were raised in a room with a 12:12 light–dark cycle at a temperature of 25 ± 2 °C and lights on from 6:00 a.m. to 6:00 p.m. Throughout the experiment, the animals had unlimited access to rodent diets (C.P. Company, Bangkok, Thailand) and reverse osmosis (RO) water. The animal ethics committee, Faculty of Medicine, Chiang Mai University, Thailand, approved the procedures for using experimental animals under protocol 7/2555, in accordance with the Code of Ethics for the Use of Animals in Scientific Work (National Research Council of Thailand, 2006). After each feeding test, the animals were euthanized by injecting a lethal dose of 50 $\mathrm{mg}\;kg^{-1}$ BW of thiopental sodium (THIOPENTAL, Biopharma, Bangkok, Thailand) intraperitoneally (i.p.). Immediately after cardiac puncture, blood was collected. Following that, the heart, spleen, thymus, liver, lung, and kidney were aseptically removed, and their weights were measured [56,57].

### 2.3. Acute Toxicity Test

During the acute toxicity test, the animals were administered a single dose of SDOs at a dose of 2000 $\mathrm{mg}\;kg^{-1}$ BW. Subsequently, atypical consequences such as mortality, gastrointestinal distress, an unsteady gait, emesis, and shivering were documented. The symptoms were carefully observed within the first 24 h period. If the SDOs are toxic, the animal will die during this period. However, if no mortality occurs within the initial 24 h period, the animals will be reared as typical animals for a duration of 14 days with unrestricted access to food and water [43,44,58]. The animals’ initial and final weights were measured at the start and end of SDO feeding. An investigation was carried out to examine the histopathological, hematological, and serum parameters of internal organs and blood.

### 2.4. Chronic Toxicity Test

For the chronic toxicity test, the rats were divided into four groups (N = 6) as follows: Group 1 was the control group that received 1 mL of distilled water; Groups 2, 3, and 4 were the experimental groups, each of which was given via oral gavage 1 mL of SDOs at doses of 50, 100, and 200 $\mathrm{mg}\;kg^{-1}$ BW/day for 6 months, respectively. The body weight of the rats was measured at the beginning and the end of this study. Every month, blood glucose was measured by drawing blood from the ends of the rat tails. Blood glucose levels were then measured using glucose test strips (Accu-Chek instant, Roche Laboratories, Pharma, Mannheim, Germany [59]) in accordance with the manufacturer’s protocol and within the indicated validity period. Systolic blood pressure (SBP) was measured monthly using the tail-cuff method with a LE5001 non-invasive blood pressure meter (Panlap, Harvard Apparatus, Barcelona, Spain) [60]. For the measurement of SBP, the rats were immobilized in a heating box (PMB-1030 Preheat & Measuring Box, MUROMACHI KIKAI CO., LTD, Tokyo, Japan) at a temperature of 37 °C for a duration of 15 min. This was conducted to measure the pulse of the tail artery as a signal to initiate blood pressure measurements. A sufficient level of pulse is required for the start button to function. If there is no signal or the pulse level is too high, the start button will not work. For each rat, multiple SBP measurements were taken, and the mean of three values that were not more than 10 mmHg apart was accepted as the SBP [56,61,62]. Clinical chemistry and hematology studies were conducted at the end of this study [43,63].

### 2.5. Clinical Chemistry and Hematological Test

After the final blood sugar test, the blood was collected from the animals at the end of the experiment and sent for clinical chemistry testing, including blood urea nitrogen (BUN), creatinine, uric acid, cholesterol, high-density lipoprotein (HDL), low-density lipoprotein (LDL), triglycerides, total protein, albumin, total bilirubin, direct bilirubin, aspartate aminotransferase (AST/SGOT), alanine aminotransferase (ALT/SGPT), and alkaline phosphatase. Hematological values include hemoglobin (Hb), hematocrit (Hct), white blood cells (WBCs), neutrophils, lymphocytes, monocytes, eosinophils, basophils, atypical lymphocytes, band form, platelet count, mean corpuscular volume (MCV), mean corpuscular hemoglobin (MCH), mean corpuscular hemoglobin concentration (MCHC), and RBC morphology [58,63,64,65].

### 2.6. Histopathological Test

At the completion of the experiment, organs such as the heart, lung, liver, kidney, adrenal gland, prostate gland, testis, and ovary were removed, weighed, and fixed in 10% formalin. The tissue was cut into small pieces of less than 3 μm thick using a microtome and prepared accordingly. The tissue and cells were then stained with hematoxylin and eosin (H&E) for further histopathological investigation [63]. The examination of organ tissues was conducted by a pathologist in accordance with the Standardized System of Nomenclature and Diagnostic Criteria (SSNDC) guidelines, as recommended by the Best Practices Guideline for Toxicologic Histopathology [66,67].

### 2.7. Statistical Analysis

Data for the acute toxicity test were analyzed using the independent-samples *t*-test. For the chronic toxicity test, statistical data were analyzed using one-way analysis of variance (ANOVA). Differences between groups were compared using the LSD test in the SPSS 16.0 program for Windows (SPSS Inc., Chicago, IL, USA) to test statistical differences and confidence level values at the *p* < 0.05 level.

## 3. Results and Discussion

### 3.1. Acute Toxicity Test

#### 3.1.1. Body and Internal Organ Weight

There were no fatalities among the mice or rats that were administered SDOs at a dosage of 2000 $\mathrm{mg}\;kg^{-1}$ BW within the initial 24 h period. No abnormal side effects were observed, including mortality, gastrointestinal distress, unsteady gait, emesis, or shivering. The animals were maintained under identical conditions to the control groups for an additional 14-day period. Based on the absence of fatalities and any indications of distress, it can be concluded with confidence that the ${LD}_{50}$ for SDOs exceeds 2000 $\mathrm{mg}\;kg^{-1}$ BW for single ingestion. The rodents were weighed on the 14th day and subsequently euthanized using thiopental sodium. The blood and internal organs were promptly collected. Table 2 shows that there was no statistically significant difference in the body weights of the female mice in the group that received a single dosage of SDOs at a dose of 2000 $\mathrm{mg}\;kg^{-1}$ BW compared to the control group. Like the ICR mice, the body weight of the rats treated with SDOs did not show any significant differences compared to the control group. Table 2 displays the weights of internal organs and their relative proportions. There were no notable variations in the weights of internal organs among female mice. In contrast to the mice, the rats that received SDOs exhibited a notable decrease in ovary weights compared to the control group. The internal organ weight and relative organ weight of ICR mice were similar to a previous study [68]. Toxicity is a concern when the effects on a specific organ are similar and significant for the two species [69]. This acute oral toxicity study of SDOs from yellow silk found no evidence.

Organ weight is an important indicator of an animal’s physiological and pathologic state [70], and it can be interpreted alongside findings from gross pathology, clinical pathology, and histopathology to understand the effects of a test substance [71]. However, it is not always the case that detectable organ weight changes directly correlate with treatment or signal adverse effects. The Society of Toxicologic Pathology recommends that pathologists evaluate organ weight data due to their expertise in correlating changes with clinical pathology, gross, and microscopic findings [71]. To determine whether changes in organ weight are disproportionate, scientists use relative organ weights to normalize data and ease comparisons among other animals [72]. Research on the safety of methanolic and aqueous extracts of *Ocimum sanctum* found that the adrenal gland weight increased in female rats but no other changes in gross or microscopic findings [73]. In 2023, Murwanti et al. found a significant difference in kidney weights in animals that were given herbal mixtures of *Piper crocatum* Ruiz and Pav., *Typhonium flagelliforme* (Lodd.) Blume, and *Phyllanthus niruri* L., but these differences were considered incidental and did not correlate with the product’s administration [74]. Similarly, female rats receiving 2000 $\mathrm{mg}\;kg^{-1}$ BW SDOs had smaller ovaries. This suggests that the ovarian weight change may be incidental and unrelated to treatment due to the insignificance in the microscopic observation.

#### 3.1.2. Clinical Chemistry

Table 3 displays the clinical chemistry results obtained from ICR mice and Wistar rats. The mice treated with SDOs exhibited elevated cholesterol and alkaline phosphatase levels (U/L) while demonstrating decreased levels of alanine aminotransferase (ALT/SGPT) compared to the control group. The rats treated with SDOs exhibited a significant elevation in cholesterol levels. Nevertheless, there were no significant differences observed in alkaline phosphatase and ALT/SGPT levels. In comparison to the control rats, the group treated with SDOs exhibited significantly elevated levels of uric acid, HDL, total protein, albumin, and globulin, in addition to cholesterol. The clinical chemistry values of the mice and rats remained within the normal range [75,76,77,78], suggesting no safety concerns regarding the toxicity of SDO consumption at a high dose.

The administration of SDOs at a high dose of 2000 $\mathrm{mg}\;kg^{-1}$ BW has been linked to an increase in uric acid levels in the blood, particularly in rats. Oligopeptides enter the body and undergo intracellular enzymatic degradation into amino acids [79] which leads to the production of purines [80]. The body naturally breaks down purines and metabolizes them into uric acid as a waste product [81]. Excessive peptide breakdown could result in purine overload, leading to elevated uric acid production. However, mice, being smaller and having a higher metabolic rate [82], may be able to process and clear metabolic byproducts like uric acid more effectively. This suggests that the elevated uric acid levels in rats may be linked to their slower metabolism, resulting in a more significant accumulation of uric acid when exposed to high peptide loads. However, uric acid levels still remained within the normal physiological range [83].

Our study demonstrated a significant increase in total cholesterol and HDL levels relative to the control group. The rise in HDL in conjunction with total cholesterol indicates that the atherogenic index (total cholesterol/HDL ratio) remains within a low-risk range (total cholesterol/HDL ratio < 5) [84,85,86]. A lower atherogenic index correlates with a decreased probability of cholesterol accumulation in the arteries, thus reducing the risk of atherosclerosis [87]. This may facilitate the transfer of excess cholesterol from peripheral tissues back to the liver for metabolism and excretion. Reverse cholesterol transport is an essential mechanism for the maintenance of cholesterol homeostasis [84]. Our findings indicate that SDOs may enhance this mechanism by increasing the HDL concentration. The proportional increase in HDL, despite the rise in total cholesterol, indicates that HDL plays a significant role in regulating cholesterol levels, thus mitigating the adverse effects typically associated with hypercholesterolemia. The results from a previous study indicated that sericin increased HDL levels in hypercholesterolemic rats via mitochondria function by enhancing its structure in the heart and liver [88].

Under normal conditions, HDL efficiently reverses cholesterol transport from peripheral tissues [84]. HDL’s antioxidant activity in mitochondria protects LDL from oxidation, maintaining cardiovascular health. LDL controls cholesterol transfer to cells, reducing artery buildup [89]. HDL production and lipid metabolism depend on liver mitochondria, while cardiac mitochondria generate energy and optimize lipid utilization [90]. Since mitochondria generate reactive oxygen species (ROS), which damage them, HDL antioxidant activity is linked to mitochondrial health [91]. High oxidative stress may impair mitochondria [92], reducing the cell’s antioxidant defenses and resulting in the increased production of ROS. This deficit may lower the cell’s antioxidant defenses and increase ROS generation. HDL and LDL can be oxidized by ROS in high amounts. Oxidized LDL is more likely to build up in artery walls, promoting plaque development and atherosclerosis, while oxidized HDL loses its function in reverse cholesterol transfer and cardiovascular health [93,94]. SDOs may decrease the ROS level via their intrinsic antioxidant properties, similar to sericin but with higher efficacy, as Wu et al. (2008) showed that enzymatically hydrolyzed bioactive peptides of lower molecular weight from sericin exhibit superior antioxidant activities compared to intact sericin [95].

Total protein and albumin levels are crucial indicators of liver and kidney function, with decreased levels indicating hepatic and renal dysfunction leading to protein loss in the urine [96]. The increased total protein in rats receiving 2000 $\mathrm{mg}\;kg^{-1}$ BW SDOs likely reflects enhanced protein synthesis, possibly due to the amino acid and peptide content of SDOs. These bioactive components may support more efficient protein metabolism, contributing to the rise in circulating protein levels [97]. Albumin levels, primarily regulated by the liver, increased significantly, indicating improved liver function [98]. SDOs may help protect the liver by increasing the production of albumin [99], shielding liver cells from oxidative stress [100], and making protein synthesis more efficient in the liver [101]. High levels of globulins, which are important for the immune system, may show that SDOs have an immunomodulatory effect, which is in line with what we already know about sericin and SDOs’ role in improving immune functions [102,103,104].

The reduction in ALT levels following 14 days of SDO treatment suggests that SDOs may have hepatoprotective effects, preventing or minimizing liver damage [105]. The animals did not show any abnormalities. The ratios of alanine aminotransferase (ALT/SGPT) and aspartate aminotransferase (AST/SGOT) remained constant in both mice and rats. This is consistent with the potential antioxidant and anti-inflammatory properties of SDOs [95,106], which may reduce oxidative stress and inflammation in liver tissues, leading to less ALT leakage into the bloodstream [105]. The reduced levels in this case suggest that a high dose of SDOs does not induce significant liver cell damage over the acute phase of 14 days. Elevated ALP levels often indicate biliary obstruction or cholestasis [107], suggesting that SDOs may have impacted bile secretion or flow, potentially leading to mild biliary stress or altered bile duct function. However, the histopathological results did not show any obstruction in the liver. Elevated levels may also suggest increased osteoblastic activity, possibly linked to the SDOs’ effects on bone metabolism [108].

#### 3.1.3. Hematology and Histopathology

The mice, on the other hand, had significantly higher white blood cell (WBC) and lymphocyte levels but significantly lower neutrophils and basophils than in the control group. In the group of the rats given SDOs, white blood cells (WBCs) were significantly higher, but eosinophil levels were significantly lower than those in the control group (Table 4). The histopathology of the internal organs from the animals, including the heart, lung, liver, kidney, etc., did not reveal any lesions in either the control or the SDO-treated groups, as reported by the clinical pathologist’s examination. According to the acute toxicity results, SDOs showed no safety concerns for one-time high-dose consumption.

This study found that SDOs have an immunomodulatory effect on both mice and rats, causing changes in their hematological profiles similar to a study by Jantaruk et al. (2015) [104]. These changes include an increase in white blood cell (WBC) counts and specific alterations in different leukocyte subpopulations. These changes remain within normal physiological ranges [75,109], indicating that SDOs have mild immunomodulatory effects without inducing toxicity or overwhelming the immune system [103,104]. The elevated WBC counts suggest that SDOs have a stimulative effect on the immune system, increasing overall immune vigilance without provoking excessive reactions [110]. The body’s adaptation to high doses prevents potential immune overstimulation, potentially leading to chronic inflammation or immune dysregulation [111].

A decrease in neutrophils that are responsible for infection and inflammation suggests that SDOs help manage excessive inflammation while preserving the body’s ability to respond. Basophils, involved in allergic and hypersensitivity responses, also decreased, suggesting that SDOs may attenuate these responses while maintaining normal immune function [112]. Eosinophils, involved in allergy responses and parasite defense [113], decreased, suggesting that SDOs can modulate immunological hypersensitivity or allergic reactions without compromising the body’s ability to combat infections or allergens [114]. This suggests that SDOs contribute to immune health and systemic balance, enhancing protection without overtaxing the body’s defense mechanisms [103,104].

This study focused on the acute effects. Exposure to SDOs over an extended period of time should produce more detailed results. According to the various test results, the acute toxicity studies found that administering SDOs at a dose of 2000 $\mathrm{mg}\;kg^{-1}$ BW was safe.

### 3.2. Chronic Toxicity Test

#### 3.2.1. Body and Internal Organ Weight

In comparison to the control group, the weight differences between male and female rats in the groups treated with different doses of SDOs were not statistically significant. The weight of the male rats increased by 260 g from 258 g at the beginning to 518 g at the end. In a similar fashion, the female rats’ weight increased by 70 g, from 213 g to 283 g. At 8 months of age, the weight gain of the rats and mice fell within the expected range [78,115]. Male rats administered SDOs at a dose of 200 $\mathrm{mg}\;kg^{-1}$ BW exhibited a significant rise in prostate gland weights compared to the control group, as illustrated in Table 5. Female rats administered doses of SDOs at 50 and 100 $\mathrm{mg}\;kg^{-1}$ BW exhibited an increased ovary weight compared to the control group. However, this effect was not observed in rats given a dose of 200 $\mathrm{mg}\;kg^{-1}$ BW SDOs. The lung weights of the group given SDOs at a dose of 200 $\mathrm{mg}\;kg^{-1}$ BW were found to be significantly higher than those of the control group. The observed variations in internal organ weight fell within the expected range [116].

During chronic toxicity tests, neither male nor female rats showed any abnormalities from histopathological and clinical chemistry studies when compared to the control group. SDOs produced from yellow silk cocoons have been shown in this study to be safe at all tested dosages. Although the rats’ lung weights were heavier, they remained within the normal range, suggesting adaptive physiological changes rather than toxicity [117]. Long-term SDO exposure may have caused the lungs to develop in response to metabolic or structural changes without causing organ dysfunction or impairment [118].

Sericin has been shown to elevate serum testosterone levels and modulate testicular growth hormone receptors [119], thereby stimulating the prostate gland [120]. Increased testosterone levels are associated with greater prostate weight and size. Sericin may influence ovarian function and growth via hormonal modulation, particularly through the growth hormone/IGF-1 (GH/IGF-1) axis. IGF-1 plays a crucial role in ovarian physiology, working in conjunction with gonadotropins, especially follicle-stimulating hormone (FSH) and luteinizing hormone (LH), to regulate follicular development and steroidogenesis [121]. The GH/IGF-1 axis increases the sensitivity of ovarian follicles to gonadotropins, thereby enhancing the responsiveness of the ovaries to hormonal signals. IGF-1 enhances the proliferation of granulosa cells in follicles, thereby facilitating follicular growth and maturation [122]. As the number of mature follicles increases, the ovary gains mass, resulting in an overall increase in ovarian weight. Increased levels of IGF-1 enhance the sensitivity of ovarian follicles to FSH and LH, thereby facilitating more vigorous follicle development and steroid hormone synthesis. This results in heightened metabolic and proliferative activity in the ovaries, which contributes to increased tissue mass and, subsequently, an elevation in ovarian weight [123]. Following SDO administration, our results showed a similar trend, with an increase in prostate gland and ovarian weight in male and female rats, respectively.

#### 3.2.2. Clinical Chemistry

Table 6 demonstrates that male rats in the group that received SDOs at a dose of 200 $\mathrm{mg}\;kg^{-1}$ BW exhibited lower cholesterol levels but higher levels of uric acid compared to the control group. The male rats in both groups that received doses of 100 and 200 $\mathrm{mg}\;kg^{-1}$ BW exhibited significantly elevated albumin levels compared to the control group. The levels of alkaline phosphatase (U/L) were found to be significantly elevated in female rats administered SDOs at a dose of 200 $\mathrm{mg}\;kg^{-1}$ BW, in comparison to the control group.

#### 3.2.3. Hematology

Table 7 indicates that male rats administered SDOs at dosages of 100 and 200 $\mathrm{mg}\;kg^{-1}$ BW had elevated neutrophil counts compared to the control group, whereas their lymphocyte counts were significantly reduced. The administration of SDOs at a dose of 200 $\mathrm{mg}\;kg^{-1}$ BW showed an opposite effect on platelet count in male and female rats. In comparison to the control group, the platelet counts of the male rats significantly decreased, whereas those of the female rats significantly increased.

After 6 months, male rats showed an increase in neutrophil counts contrary to the decrease observed in the acute toxicity test. This suggests that chronic exposure to SDOs may have a stimulating effect on innate immune cells [124], such as neutrophils, potentially enhancing the body’s readiness to respond to infections or inflammation in the long term [125]. The lymphocyte levels decreased, which is also the opposite of the acute response. The high-dose SDO exposure might temporarily activate the immune response, causing the acute increase in lymphocytes. However, with long-term exposure, the decrease in lymphocytes might reflect a balancing effect where chronic SDO intake modulates immune responses maintaining homeostasis with prolonged exposure. Male rats showed a decrease in platelet count, which could indicate that SDOs may have a sex-specific effect on thrombopoiesis (platelet production) [126,127]. However, the changes in platelet counts were still within the normal range [75,109].

#### 3.2.4. Blood Sugar Levels

During the initial month, male rats that received SDOs at dosages of 50 and 100 $\mathrm{mg}\;kg^{-1}$ BW exhibited significantly lower blood sugar levels compared to the control group. There were no notable changes observed during this study, except for the sixth month in the group that received a dose of 200 $\mathrm{mg}\;kg^{-1}$ BW of SDOs, as indicated in Table 8. In the initial month of this study, female rats administered SDOs at a dose of 200 $\mathrm{mg}\;kg^{-1}$ BW had significantly lower blood sugar levels compared to the control group. However, there was no notable difference in blood sugar levels observed between the two groups over the course of this study.

SDOs may affect blood sugar regulation in both diabetic and normoglycemic rats by inhibiting the enzymes α-amylase, α-glucosidase, and dipeptidyl peptidase-4 (DPP-4) [5]. However, there are notable differences in the responses of these two groups associated with the degree of enzyme inhibition and its effect on blood glucose levels. The inhibition of α-amylase and α-glucosidase is especially important in diabetic rats, characterized by impaired insulin sensitivity and hyperglycemia. The slower digestion of carbohydrates results in less glucose being absorbed, which considerably lowers postprandial blood sugar spikes [128]. Furthermore, DPP-4 inhibition is crucial as it increases the availability of GLP-1 and glucose-dependent insulinotropic polypeptide (GIP), thereby enhancing insulin secretion [129]. This makes up for the impaired insulin production or efficacy that is frequently seen in diabetic conditions [130].

In normoglycemic rats, where blood glucose regulation is already functioning optimally, the effect of SDOs was less pronounced. The lower basal enzyme activity of α-amylase and α-glucosidase suggests less carbohydrate breakdown and a more controlled insulin response [131]. When SDOs are administered, they may still inhibit these enzymes, but the degree of the reduction in blood glucose levels is less substantial compared to diabetic rats because the system is not overburdened with excess glucose or insulin resistance [132]. SDOs still lower blood sugar in normoglycemic rats but require a higher dose to achieve noticeable effects. This is due to the fact that, in a non-diabetic state, blood sugar homeostasis is tightly regulated, maintaining a healthy range [132,133]. The body’s mechanisms for balancing insulin, glucose uptake, and hormone regulation are already efficient, so the impact of SDO inhibition on these enzymes is comparatively smaller. This indicates that SDOs may have a dose-dependent effect on glucose metabolism, more effective in pathological states like diabetes but still capable of modulating blood sugar in normal conditions to a lesser degree. The availability and function of incretin hormones, such as GLP-1 and GIP, are affected by DPP-4 inhibition [129]. While GLP-1 also suppresses glucagon to regulate blood sugar, GIP stimulates glucagon release from the pancreas and hepatic glucose production [134]. When it comes to managing blood sugar levels, these two hormones might have opposing effects. Upon the administration of SDOs, this dual effect may help sustain glucose homeostasis in normoglycemic rats, with blood glucose levels that are already in balance. The simultaneous elevation of insulin and glucagon may assist the body in regulating blood glucose levels, thereby preventing substantial fluctuations [132]. This may explain why SDOs still lower blood sugar in normoglycemic rats but with a less dramatic effect than in diabetic rats, where insulin activity is more critically impaired.

#### 3.2.5. Blood Pressure Levels

Throughout this study, all groups receiving SDOs demonstrated a significant reduction in the systolic blood pressure (SBP) of male rats (Figure 1) when compared to the control group. During the testing period, the groups that received SDO treatment consistently maintained systolic blood pressure (SBP) values around 120 mmHg. In contrast, the control group experienced an increase in SBP to over 140 mmHg by the end of this study. Figure 2 depicts the systolic blood pressure (SBP) of female rats. During the second month of the treatment, the group that received SDOs at a dose of 200 $\mathrm{mg}\;kg^{-1}$ BW began to exhibit a significant decrease in blood pressure compared to the control group. This decrease was maintained within the range of 110 to 120 mmHg for the duration of this study. Throughout the fourth to sixth month, the groups treated with SDOs consistently exhibited significantly lower systolic blood pressure (SBP) compared to the control group. The rats administered SDOs at various doses exhibited similar systolic blood pressure (SBP) levels after six months, approximately 115 mmHg. Significant reductions in systolic blood pressure (SBP) were observed in the second month with high doses of SDOs, such as 200 $\mathrm{mg}\;kg^{-1}$ BW. Interestingly, lower doses of 50 $\mathrm{mg}\;kg^{-1}$ BW also demonstrated comparable effects when administered over an extended duration. The doses of SDOs displayed a consistent pattern: as the SBP decreased, it eventually stabilized at around 115 mmHg for male and female rats. When examining the rats’ lifespan, which is around two years, it is noteworthy that consuming SDOs for six months would account for a quarter of their total lifespan. It is important to mention that the systolic blood pressure (SBP) of male and female rats in the control groups showed a gradual increase as they aged, eventually exceeding 140 mmHg by the end of this study.

In the second month of exposure to all SDOs doses, male and female rats had lower blood pressure than the control group. In the fourth month of the trial, the rats’ blood pressure stabilized and stayed within range for reproductively mature rats [135]. Animals’ blood pressure rises with age [136]. Our research found that rats given all dosages of SDOs had lower blood pressure, maintaining healthy values throughout the trial. We found that SDOs had an antihypertensive impact from the 8th week of administration which lasts up to 6 months. This length is approximately 25% of rats’ typical lifetime, highlighting SDOs’ potential long-term blood pressure control benefits. The blood pressure decreased dose-dependently, although all treatment groups plateaued by the 4th month. The SDO dosage must be optimized to provide therapeutic benefits because of the dose–response relationship. SDOs reduced blood pressure in both male and female rats, making them a prospective antihypertensive therapeutic. SDOs lower blood pressure via the endothelial-dependent NO/sGC/cGMP pathway vasorelaxation, directly affecting vascular smooth muscle [6] and inhibiting ACE in the bloodstream [7].

Nitric oxide (NO) is a signaling molecule that is produced by the endothelial cells lining the blood vessels. NO and soluble guanylate cyclase (sGC) play an important role in regulating blood pressure. The two molecules are involved in blood pressure lowering by inducing vasodilation (the widening of blood vessels). NO achieves this by diffusing into the smooth muscle cells of blood vessels and stimulating sGC [137]. sGC is an enzyme found in the smooth muscle cells of blood vessels. When NO binds to sGC, it activates the enzyme. The activated sGC then converts guanosine triphosphate (GTP) to cyclic guanosine monophosphate (cGMP). cGMP prevents calcium influx, subsequently causing smooth muscle to relax, leading to vasodilation and a reduction in blood pressure. Through this process, NO and sGC work together to help regulate blood pressure by promoting the relaxation of blood vessels. Any disruption in the NO/sGC pathway can lead to issues with blood pressure regulation, potentially contributing to conditions like hypertension [138]. Therefore, SDOs may exert their hypotensive effect potentially by enhancing the production of NO. An in vitro study by Aramwit et al. (2009) [139] showed that silk sericin could enhance cellular NO production.

A previous study reported the vasorelaxation effect of SDOs in aortic endothelium-denuded rings, suggesting that SDOs act directly on vascular smooth muscle cells as well as through endothelial cells [6]. The effect may result from the opening of $K^{+}$ channels or the blockage of ${Ca}^{2+}$ channels, which can reduce calcium influx and facilitate muscle relaxation. In vascular smooth muscle cells, hyperpolarization caused by $K^{+}$ channel opening reduces the probability of ${Ca}^{2+}$ channel opening, reducing calcium influx and facilitating muscle relaxation. The blockage of ${Ca}^{2+}$ channels directly inhibits calcium influx into smooth muscle cells, leading to muscle relaxation and vasodilation. SDOs also significantly reduced the contractions of vascular smooth muscle cells triggered by phenylephrine which stimulate the release of calcium ions from intracellular stores (sarcoplasmic reticulum) via an IP3-dependent pathway [140]. This suggests that SDOs may act as calcium channel blockers or influence intracellular signaling pathways related to calcium release, ultimately disrupting the calcium release mechanism.

Furthermore, earlier in vitro research on the dual role of SDOs indicated their capacity to inhibit dipeptidyl peptidase-4 (DPP4) and angiotensin-converting enzyme (ACE) [7,27]. The dosage necessary for DPP4 inhibition was considerably greater than that for ACE inhibition; however, both demonstrated significant efficacy. After degradation by blood plasmin, SDOs demonstrated increased ACE inhibitory activity in vitro, indicating improved efficacy upon metabolism in the body. The inhibitory effect on ACE at low concentrations likely contributes to the observed reduction in blood pressure in vivo (unpublished data).

Additionally, another study indicated that SDOs exhibit hypolipidemic effects by lowering non-high-density lipoprotein (non-HDL), triglyceride, and total serum cholesterol levels [141]. The hypolipidemic effect is significant, suggesting that SDOs may aid in blood pressure regulation while improving lipid profiles, thereby providing an overall cardiovascular protective advantage. Our findings are consistent with prior studies demonstrating the cardiovascular health benefits of bioactive peptides derived from plant, milk, and marine sources [142,143,144]. SDOs are distinguished by their prolonged effectiveness and short duration of effects. The capacity to reduce blood pressure from the 8th week to 6 months while sustaining a healthy athletic range (120 mmHg) indicates that SDOs may offer early therapeutic advantages, potentially mitigating the risk of long-term hypertension-related complications. Investigation into the effectiveness of SDOs for lowering blood pressure in hypertensive rats is necessary.

According to the findings of this study, SDOs exhibited neither acute toxicity at a fixed dose of 2000 $\mathrm{mg}\;kg^{-1}$ BW nor chronic toxicity at 50, 100, and 200 $\mathrm{mg}\;kg^{-1}$ BW in both male and female Wistar rats. Through the integration of regular assessments of vital signs, such as blood pressure and blood sugar, during the 6-month chronic toxicity study, we were able to offer a more comprehensive understanding of the health and well-being of the rats, both throughout their lifespan and as they aged. The duration of chronic toxicity often extends up to six months, representing around 25% of the typical lifespan of the experimental rats. The data obtained from live and aging rats, especially in terms of SBP and blood glucose levels, may provide insights into the influence of the oral administration of SDOs at low dosages and indicate that SDOs could potentially contribute to general health and fitness related to longevity. Previous investigations on SDOs have demonstrated a variety of bifunctionalities and mechanisms of action over the past few decades. Based on this information, we propose that the inherent strong antioxidant activity of SDOs, their inhibitory effects on pancreatic and ACE enzymes, and their enzymatic resistance in the gastrointestinal tract and bloodstream provide benefits for their transport into the circulatory system and their functional expression at various receptive sites in multiple internal organs, including mitochondria, thereby opening up novel research approaches. Measuring vital signs in live and aging laboratory animals could be labor-intensive and time-consuming and necessitates appropriate facilities, especially when evaluating a substantial cohort of animals.

## 4. Conclusions

Rats and mice, the two experimental female animal species tested for acute toxicity, did not exhibit signs or symptoms of SDOs from yellow silk cocoons. Histopathological examination of the internal organs revealed no abnormalities in any of the animals. This research also found no significant effects on clinical chemistry, with the exception of uric acid, cholesterol, HDL, albumin, and globulin levels. This could potentially be due to the high dosages of SDOs, which could potentially damage liver function. Male and female rats showed no abnormalities throughout the chronic toxicity test. Moreover, all SDO-treated rats had a lowered mean SBP from 130 mmHg which remained in the young healthy range of 110–115 mmHg after four months, while the control rats continued to have increased SBP as they aged which reached over 140 mmHg at the end of the study period, suggesting hypertensive onset. The SDO-treated rats had slightly lower blood sugar levels than the control rats, but there were no significant differences. The chronic toxicity, along with periodic checks of SBP and blood sugar levels in live rats as they aged, provided useful information on the hypotensive effects of SDOs on aging rats. This could lead to an investigation on hypertensive rats, but it could be time-consuming and labor-demanding and require specific measurement equipment and facilities.

## Figures and Tables

**Figure 1 foods-13-03505-f001:**
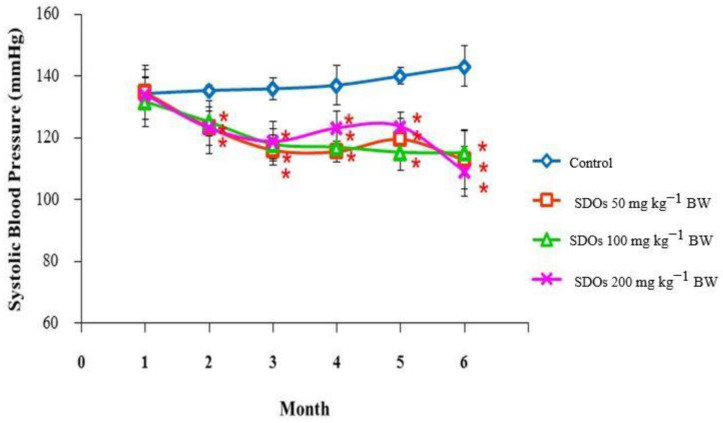
Comparison of the systolic blood pressure of control male Wistar rats with those receiving different doses of SDOs for 6 months. Values are expressed as mean ± SD. * *p* < 0.05; six rats per group.

**Figure 2 foods-13-03505-f002:**
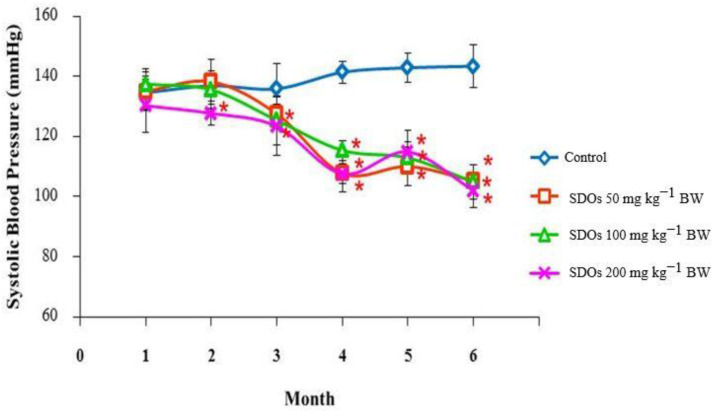
Comparison of the systolic blood pressure of control female Wistar rats with those receiving different doses of SDOs for 6 months. Values are expressed as mean ± SD. * *p* < 0.05; six rats per group.

**Table 1 foods-13-03505-t001:** Specifications of SDOs from yellow silk cocoons.

Items	Specification
Source	Yellow silk cocoon
Molecular weight	<5 kDa
Color	Wheaten powder
pI	pH 10–12
% moisture	Approx. 5%
Purity	95–99% (Kjeldahl method using 6.25 conversion factor) [55]
Trace metals:	
Pb max	0.045 $\mathrm{mg}\;kg^{-1}$
Cd max	0.003 $\mathrm{mg}\;kg^{-1}$
Hg max	0.007 $\mathrm{mg}\;kg^{-1}$
As max	0.035 $\mathrm{mg}\;kg^{-1}$
Zn max	14.333 $\mathrm{mg}\;kg^{-1}$
Ni max	0.518 $\mathrm{mg}\;kg^{-1}$
Microbial standards:	
Total plate counts	85 CFU/g
Yeast and Molds	20 CFU/g
*E. coli*	<3.0 MPN/g
*Salmonella* spp.	Not detected
*Staphylococcus aureus*	Not detected
Coliforms (MPN)	<3.0 MPN/g
*Listeria monocytogenes*	Not Detected
Pesticide residue:	
Carbamate group	Not detected
Organochlorine group	Not detected
Organophosphate group	Not detected
Pyrethroid group	Not detected

**Table 2 foods-13-03505-t002:** Comparison of absolute and relative (%) internal organ weights of control ICR mice and Wistar rats with those receiving SDOs at a dose of 2000 $\mathrm{mg}\;kg^{-1}$ BW after 14 days of observation.

Parameters	Female ICR Mice		Female Wistar Rats	
	Control	$\mathrm{SDOs}\;2000\;\mathrm{mg}\;kg^{-1}$ BW	Control	$\mathrm{SDOs}\;2000\;\mathrm{mg}\;kg^{-1}$ BW
Body weight (g) Heart (g) %	35.67 ± 3.67	35.67 ± 2.88	236.33 ± 4.08	234.50 ± 4.76
	0.13 ± 0.01	0.13 ± 0.01	0.78 ± 0.03	0.81 ± 0.06
	0.36 ± 0.04	0.36 ± 0.02	0.24 ± 0.03	0.24 ± 0.01
Lung (g) %	0.17 ± 0.01	0.17 ± 0.01	1.00 ± 0.06	0.97 ± 0.03
	0.49 ± 0.05	0.47 ± 0.05	0.31 ± 0.05	0.28 ± 0.01
Liver (g) %	1.25 ± 0.10	1.26 ± 0.06	7.42 ± 0.15	7.18 ± 0.62
	3.52 ± 0.45	3.53 ± 0.18	2.27 ± 0.32	2.08 ± 0.15
Kidney (g) %	0.38 ± 0.01	0.38 ± 0.01	1.59 ± 0.14	1.55 ± 0.05
	1.09 ± 0.10	1.07 ± 0.07	0.49 ± 0.10	0.45 ± 0.02
Adrenal gland (g) %	0.01 ± 0.00	0.01 ± 0.00	0.10 ± 0.01	0.09 ± 0.01
	0.03 ± 0.00	0.03 ± 0.01	0.03 ± 0.00	0.03 ± 0.00
Ovary (g) %	0.02 ± 0.00	0.02 ± 0.00	0.21 ± 0.02	0.15 ± 0.02 *
	0.06 ± 0.01	0.06 ± 0.01	0.06 ± 0.02	0.05 ± 0.00 *

Values are expressed as mean ± SD. * *p* < 0.05; six rats per group. Relative (%) = ratio of organ weight to body weight × 100.

**Table 3 foods-13-03505-t003:** Comparison of clinical chemistry test of control ICR mice and Wistar rats with those receiving SDOs at a dose of 2000 $\mathrm{mg}\;kg^{-1}$ BW after 14 days of observation.

Parameters	Female ICR Mice		Female Wistar Rats	
	Control	$\mathrm{SDOs}\;2000\;\mathrm{mg}\;kg^{-1}$ BW	Control	$\mathrm{SDOs}\;2000\;\mathrm{mg}\;kg^{-1}$ BW
Glucose (mg/dL)	98.00 ± 4.86	100.00 ± 4.34	105.67 ± 5.13	97.83 ± 8.89
BUN (mg/dL)	22.13 ± 1.32	23.47 ± 1.81	28.28 ± 3.00	32.22 ± 8.98
Creatinine (mg/dL)	0.29 ± 0.04	0.31 ± 0.05	0.64 ± 0.08	0.62 ± 0.05
Uric acid (mg/dL)	3.68 ± 0.18	3.55 ± 0.12	1.46 ± 0.15	1.84 ± 0.23 *
Total cholesterol (mg/dL)	89.83 ± 4.17	99.67 ± 3.72 *	49.02 ± 3.89	66.48 ± 7.21 *
HDL (mg/dL)	-	-	26.85 ± 2.66	36.48 ± 3.73 *
Triglyceride (mg/dL)	-	-	57.67 ± 16.72	64.50 ± 33.58
Total protein (g/dL)	-	-	5.77 ± 0.29	6.36 ± 0.32 *
Albumin (g/dL)	-	-	1.76 ± 0.11	1.97 ± 0.12 *
Globulin (g/dL)	-	-	4.00 ± 0.19	4.39 ± 0.21 *
Total bilirubin (mg/dL)	-	-	0.72 ± 0.11	0.82 ± 0.16
Direct bilirubin (mg/dL)	-	-	0.09 ± 0.03	0.12 ± 0.04
AST/SGOT (U/L)	95.17 ± 3.49	93.50 ± 5.68	67.67 ± 8.69	66.33 ± 6.25
ALT/SGPT (U/L)	22.50 ± 1.87	17.50 ± 1.87 *	22.50 ± 3.62	25.00 ± 2.61
Alkaline phosphatase (U/L)	82.33 ± 10.33	98.00 ± 8.00 *	106.83 ± 12.64	103.83 ± 13.47

Values are expressed as mean ± SD. * *p* < 0.05; six rats per group.

**Table 4 foods-13-03505-t004:** Comparison of hematology test of control ICR mice and Wistar rats with those receiving SDOs at a dose of 2000 $\mathrm{mg}\;kg^{-1}$ BW after 14 days of observation.

Parameters	Female ICR Mice		Female Wistar Rats	
	Control	$\mathrm{SDOs}\;2000\;\mathrm{mg}\;kg^{-1}$ BW	Control	$\mathrm{SDOs}\;2000\;\mathrm{mg}\;kg^{-1}$ BW
Hemoglobin (g/dL)	13.70 ± 1.26	13.97 ± 0.77	13.90 ± 2.91	15.48 ± 0.70
Hematocrit (%)	41.75 ± 4.29	44.12 ± 4.41	39.45 ± 8.96	44.17 ± 2.02
WBC (×10^3^ cell/µL)	0.88 ± 0.29	1.30 ± 0.32 *	1.97 ± 0.27	3.53 ± 0.34 *
Neutrophil (%)	24.50 ± 5.73	17.63 ± 3.64 *	10.10 ± 2.41	8.75 ± 2.35
Lymphocyte (%)	66.65 ± 5.79	76.28 ± 4.02 *	87.68 ± 2.85	89.52 ± 2.32
Monocyte (%)	0.68 ± 0.49	0.65 ± 0.23	0.62 ± 0.42	0.72 ± 0.36
Eosinophil (%)	2.78 ± 1.36	1.87 ± 0.94	0.43 ± 0.14	0.15 ± 0.05 *
Basophil (%)	5.38 ± 0.84	3.57 ± 0.77 *	1.17 ± 0.23	0.87 ± 0.39
Platelet count (×10^3^ cell/µL)	680.33 ± 109.35	622.50 ± 91.20	663.17 ± 33.36	650.17 ± 46.90
MCV (fL)	52.00 ± 1.10	53.17 ± 1.47	58.33 ± 1.51	58.50 ± 1.38
MCH (pg)	17.00 ± 0.40	17.20 ± 0.42	20.72 ± 1.09	20.42 ± 0.63
MCHC (%)	32.80 ± 0.55	32.55 ± 0.28	35.48 ± 1.29	35.05 ± 0.37

Values are expressed as mean ± SD. * *p* < 0.05; six rats per group.

**Table 5 foods-13-03505-t005:** Comparison of absolute and relative (%) internal organ weights of control male and female Wistar rats with those receiving SDOs at various doses for 6 months.

Parameters	Male Wistar Rats				Female Wistar Rats			
		$\mathrm{SDOs}\;(\mathrm{mg}\;kg^{-1}$ BW)				$\mathrm{SDOs}\;(\mathrm{mg}\;kg^{-1}$ BW)		
	Control	50	100	200	Control	50	100	200
Body weight (g)	515.33 ± 17.64	516.17 ± 26.25	516.33 ± 21.97	518.17 ± 20.42	274.33 ± 7.28	282.83 ± 10.46	276.00 ± 5.93	281.83 ± 7.28
Heart (g) %	1.22 ± 0.07	1.21 ± 0.05	1.19 ± 0.06	1.16 ± 0.08	0.77 ± 0.07	0.83 ± 0.03	0.82 ± 0.07	0.84 ± 0.03
	0.24 ± 0.02	0.24 ± 0.02	0.23 ± 0.01	0.22 ± 0.01	0.28 ± 0.03	0.29 ± 0.01	0.30 ± 0.02	0.30 ± 0.01
Lung (g) %	1.40 ± 0.06	1.46 ± 0.06	1.45 ± 0.03	1.41 ± 0.03	0.99 ± 0.05	1.00 ± 0.11	0.97 ± 0.11	1.14 ± 0.07 *
	0.27 ± 0.01	0.28 ± 0.02	0.28 ± 0.01	0.27 ± 0.01	0.36 ± 0.02	0.35 ± 0.05	0.35 ± 0.03	0.40 ± 0.02
Liver (g) %	12.92 ± 1.04	12.56 ± 1.09	12.15 ± 0.77	12.63 ± 1.27	7.18 ± 0.28	7.38 ± 0.22	7.40 ± 0.27	7.38 ± 0.13
	2.50 ± 0.12	2.43 ± 0.16	2.35 ± 0.08	2.44 ± 0.21	2.62 ± 0.13	2.61 ± 0.11	2.68 ± 0.12	2.62 ± 0.07
Kidney (g) %	2.17 ± 0.11	2.06 ± 0.10	2.11 ± 0.07	2.21 ± 0.16	1.50 ± 0.07	1.57 ± 0.09	1.52 ± 0.08	1.53 ± 0.09
	0.42 ± 0.01	0.40 ± 0.02	0.41 ± 0.01	0.43 ± 0.03	0.55 ± 0.02	0.55 ± 0.04	0.55 ± 0.03	0.54 ± 0.02
Adrenal gland (g) %	0.07 ± 0.01	0.07 ± 0.01	0.07 ± 0.00	0.07 ± 0.01	0.07 ± 0.02	0.07 ± 0.01	0.07 ± 0.02	0.07 ± 0.01
	0.01 ± 0.00	0.01 ± 0.00	0.01 ± 0.00	0.01 ± 0.00	0.02 ± 0.01	0.03 ± 0.00	0.03 ± 0.01	0.03 ± 0.00
Spleen (g) %	0.77 ± 0.08	0.78 ± 0.10	0.86 ± 0.08	0.82 ± 0.09	0.57 ± 0.05	0.61 ± 0.08	0.65 ± 0.09	0.61 ± 0.07
	0.15 ± 0.02	0.15 ± 0.02	0.17 ± 0.01	0.16 ± 0.02	0.21 ± 0.01	0.22 ± 0.03	0.23 ± 0.03	0.22 ± 0.02
Prostate gland (g) %	0.21 ± 0.08	0.28 ± 0.06	0.28 ± 0.08	0.32 ± 0.08 *	-	-	-	-
	0.04 ± 0.01	0.05 ± 0.01	0.05 ± 0.01	0.06 ± 0.01	-	-	-	-
Seminal vesicle (g) %	0.99 ± 0.08	1.03 ± 0.09	0.94 ± 0.10	1.06 ± 0.08	-	-	-	-
	0.19 ± 0.02	0.20 ± 0.02	0.18 ± 0.02	0.20 ± 0.02	-	-	-	-
Epididymis (g) %	1.42 ± 0.05	1.44 ± 0.08	1.43 ± 0.10	1.43 ± 0.14	-	-	-	-
	0.27 ± 0.01	0.28 ± 0.02	0.28 ± 0.02	0.28 ± 0.02	-	-	-	-
Testis (g) %	3.98 ± 0.15	3.97 ± 0.12	3.96 ± 0.11	3.93 ± 0.19	-	-	-	-
	0.77 ± 0.04	0.77 ± 0.04	0.77 ± 0.03	0.76 ± 0.05	-	-	-	-
Ovary (g) %	-	-	-	-	0.09 ± 0.03	0.13 ± 0.04 *	0.13 ± 0.04 *	0.09 ± 0.02
	-	-	-	-	0.03 ± 0.01	0.05 ± 0.01	0.05 ± 0.01	0.03 ± 0.01

Values are expressed as mean ± SD. * *p* < 0.05; six rats per group. Relative (%) = ratio of organ weight to body weight × 100.

**Table 6 foods-13-03505-t006:** Comparison of clinical chemistry test of control male and female Wistar rats with those receiving SDOs at various doses for 6 months.

Parameters	Male Wistar Rats				Female Wistar Rats			
		$\mathrm{SDOs}\;(\mathrm{mg}\;kg^{-1}$ BW)				$\mathrm{SDOs}\;(\mathrm{mg}\;kg^{-1}$ BW)		
	Control	50	100	200	Control	50	100	200
Total cholesterol (mg/dL)	63.83 ± 7.78	55.83 ± 5.85	58.67 ± 8.85	53.33 ± 9.67 *	52.00 ± 7.43	55.17 ± 7.52	52.83 ± 8.98	52.83 ± 8.30
HDL (mg/dL)	50.88 ± 6.68	46.07 ± 5.90	49.12 ± 4.73	45.98 ± 5.12	41.97 ± 4.48	45.38 ± 6.10	42.78 ± 7.60	43.03 ± 5.83
Triglyceride (mg/dL)	62.33 ± 9.91	69.50 ± 18.25	54.00 ± 16.20	54.17 ± 15.54	47.00 ± 9.67	49.33 ± 8.31	42.50 ± 8.67	41.17 ± 8.52
BUN (mg/dL)	27.73 ± 1.40	28.52 ± 1.77	28.12 ± 2.30	28.68 ± 2.83	32.53 ± 4.45	38.63 ± 9.75	30.17 ± 3.51	29.83 ± 2.57
Creatinine (mg/dL)	0.76 ± 0.13	0.66 ± 0.06	0.70 ± 0.13	0.65 ± 0.17	0.57 ± 0.09	0.62 ± 0.04	0.62 ± 0.14	0.60 ± 0.08
Uric acid (mg/dL)	1.78 ± 0.19	1.95 ± 0.23	1.73 ± 0.17	2.09 ± 0.36 *	2.14 ± 0.48	1.87 ± 0.37	2.19 ± 0.26	1.79 ± 0.17
Total protein (g/dL)	6.39 ± 0.22	6.32 ± 0.36	6.45 ± 0.22	6.53 ± 0.36	6.37 ± 0.29	6.62 ± 0.51	6.58 ± 0.39	6.36 ± 0.51
Albumin (g/dL)	3.10 ± 0.08	3.15 ± 0.07	3.22 ± 0.07 *	3.26 ± 0.07 *	3.17 ± 0.18	3.15 ± 0.06	3.23 ± 0.07	3.13 ± 0.11
Total bilirubin (mg/dL)	0.59 ± 0.07	0.58 ± 0.09	0.58 ± 0.06	0.58 ± 0.04	0.64 ± 0.09	0.59 ± 0.08	0.70 ± 0.08	0.61 ± 0.08
Direct bilirubin (mg/dL)	0.08 ± 0.02	0.09 ± 0.03	0.08 ± 0.03	0.12 ± 0.04	0.14 ± 0.02	0.13 ± 0.03	0.13 ± 0.02	0.13 ± 0.03
AST/SGOT (U/L)	105.67 ± 10.19	105.33 ± 6.68	107.83 ± 10.32	105.67 ± 9.07	114.17 ± 6.59	108.50 ± 7.69	116.50 ± 9.16	113.67 ± 10.69
ALT/SGPT (U/L)	35.17 ± 6.24	34.17 ± 3.49	32.67 ± 8.50	33.67 ± 3.20	43.33 ± 8.87	41.83 ± 2.23	43.17 ± 9.13	48.00 ± 6.84
Alkaline phosphatase (U/L)	114.50 ± 15.07	113.83 ± 9.39	113.83 ± 13.17	116.67 ± 11.18	80.83 ± 5.49	86.50 ± 9.89	78.17 ± 10.26	102.33 ± 14.46 *

Values are expressed as mean ± SD. * *p* < 0.05; six rats per group.

**Table 7 foods-13-03505-t007:** Comparison of hematology test of control male and female Wistar rats with those receiving SDOs at various doses for 6 months.

Parameters	Male Wistar Rats				Female Wistar Rats			
		$\mathrm{SDOs}\;(\mathrm{mg}\;kg^{-1}$ BW)				$\mathrm{SDOs}\;(\mathrm{mg}\;kg^{-1}$ BW)		
	Control	50	100	200	Control	50	100	200
Hemoglobin (g/dL)	14.57 ± 0.77	14.82 ± 0.48	14.65 ± 0.68	14.65 ± 0.37	14.88 ± 1.14	14.50 ± 1.12	14.73 ± 0.81	14.50 ± 0.32
Hematocrit (%)	42.50 ± 2.43	42.50 ± 1.05	42.83 ± 2.64	42.67 ± 0.82	42.67 ± 3.14	42.00 ± 3.10	42.50 ± 2.07	41.50 ± 0.84
WBC (×10^3^ cell/µL)	3.67 ± 0.34	3.38 ± 0.35	3.32 ± 0.39	3.23 ± 0.48	2.43 ± 0.65	2.22 ± 0.26	2.55 ± 0.39	2.88 ± 0.28
Neutrophil (%)	14.17 ± 1.94	14.33 ± 2.42	20.83 ± 7.57 *	21.33 ± 2.34 *	23.00 ± 5.48	21.67 ± 3.27	22.83 ± 7.60	17.67 ± 5.20
Lymphocyte (%)	84.67 ± 2.25	84.67 ± 2.42	78.00 ± 7.51 *	77.50 ± 2.07 *	74.67 ± 5.01	76.83 ± 4.07	74.50 ± 8.41	81.17 ± 4.88
Monocyte (%)	1.00 ± 0.00	1.00 ± 0.00	0.67 ± 0.52	0.67 ± 0.52	1.17 ± 0.75	1.17 ± 0.41	1.50 ± 0.55	0.83 ± 0.41
Eosinophil (%)	0.17 ± 0.41	0.00 ± 0.00	0.50 ± 0.55	0.50 ± 0.55	1.00 ± 0.89	0.17 ± 0.41	1.17 ± 0.98	0.33 ± 0.52
Basophil (%)	0.00 ± 0.00	0.00 ± 0.00	0.00 ± 0.00	0.00 ± 0.00	0.17 ± 0.41	0.17 ± 0.41	0.00 ± 0.00	0.00 ± 0.00
Platelet count (×10^3^ cell/µL)	575.17 ± 38.79	560.83 ± 39.10	560.00 ± 37.21	527.50 ± 38.20 *	575.33 ± 58.81	615.50 ± 31.85	602.33 ± 38.66	645.00 ± 37.30 *
MCV (fL)	54.50 ± 1.05	54.33 ± 1.37	54.50 ± 1.87	54.50 ± 2.07	56.67 ± 1.97	56.17 ± 1.17	57.33 ± 2.42	57.00 ± 1.10
MCH (pg)	18.67 ± 0.52	18.83 ± 0.75	19.00 ± 0.63	18.83 ± 0.75	20.00 ± 1.10	19.50 ± 0.55	20.17 ± 0.75	20.17 ± 0.75
MCHC (%)	34.33 ± 0.82	34.83 ± 0.75	34.67 ± 0.82	34.33 ± 1.21	35.00 ± 0.63	34.83 ± 0.41	34.83 ± 0.41	34.83 ± 0.41

Values are expressed as mean ± SD. * *p* < 0.05; six rats per group.

**Table 8 foods-13-03505-t008:** Comparison of blood sugar levels (mg/dL) of control male and female Wistar rats with those receiving SDOs at various doses for 6 months.

Months	Male Wistar Rats				Female Wistar Rats			
		$\mathrm{SDOs}\;(\mathrm{mg}\;kg^{-1}$ BW)				$\mathrm{SDOs}\;(\mathrm{mg}\;kg^{-1}$ BW)		
	Control	50	100	200	Control	50	100	200
Initial	70.17 ± 1.94	70.33 ± 2.07	69.33 ± 2.07	70.00 ± 2.68	65.17 ± 3.49	65.00 ± 3.90	64.67 ± 3.14	64.83 ± 3.13
1	73.83 ± 5.34	64.83 ± 3.31 *	64.33 ± 4.76 *	70.33 ± 4.84	67.00 ± 4.65	64.33 ± 3.50	63.33 ± 2.34	59.50 ± 4.89 *
2	69.17 ± 4.26	67.83 ± 5.08	67.50 ± 2.35	67.33 ± 3.08	61.67 ± 3.39	65.00 ± 5.18	62.33 ± 6.50	64.17 ± 3.82
3	69.83 ± 5.15	69.50 ± 4.18	70.17 ± 3.97	69.17 ± 4.17	62.83 ± 5.00	59.50 ± 6.75	60.33 ± 2.25	61.17 ± 3.31
4	70.50 ± 3.39	68.83 ± 4.88	68.00 ± 3.90	65.67 ± 6.15	63.33 ± 4.08	67.50 ± 5.17	59.83 ± 4.75	65.83 ± 4.22
5	71.17 ± 2.64	69.83 ± 2.56	71.33 ± 3.14	68.67 ± 3.93	63.33 ± 5.57	67.33 ± 9.37	66.33 ± 4.08	65.00 ± 4.34
6	71.00 ± 3.74	67.50 ± 4.04	70.50 ± 3.73	66.00 ± 4.73 *	61.33 ± 3.39	61.33 ± 4.80	58.83 ± 6.68	66.17 ± 3.06

Values are expressed as mean ± SD. * *p* < 0.05; six rats per group.

## Data Availability

The original contributions presented in this study are included in the article. Further inquiries can be directed to the corresponding author.

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
