# Peer review of "Oral Toxicity and Hypotensive Influence of Sericin-Derived Oligopeptides (SDOs) from Yellow Silk Cocoons of Bombyx mori in Rodent Studies"

_foods, 2024, doi:10.3390/foods13213505_

Round 1
Reviewer 1 Report
Comments and Suggestions for Authors
My comments over the article:
1. The background is extensive and sufficient to understand the purpose of the research work.
2. The topic of yellow silk cocoons is widely known, and the references contain various aspects of scientific interest, including the potential toxicological effects of sericin and its oligopeptide derivatives. Therefore, this work is interesting and relevant in its contribution.
3. The toxicological evaluation places it on a level of scientific relevance, which increases interest and novelty in the research area, which is why I consider the topic relevant and important.
4. The research methodology is clearly described, following conventional protocols, and it is relevant to highlight the use of international regulations for toxicological studies, which to date seek to minimize the use and sacrifice of animals.
5. The bioethics committee authorization number for the use of animals is presented.
6. With reference to the results, the biological evaluation tables are clear and descriptive.
7. The graphs presented are legible and clearly show the description of the results in the text.
8. The discussion directly supports each result obtained, which is why the conclusions are considered to be in accordance with each experiment or evaluation carried out.
9. The writing of the work apparently does not require revision, since I felt it was fluent, without being an expert in languages.
1. The references comply with the format requested by the journal, and I found no signs of plagiarism.
1. I found several self-citations, about 13, however, they are directly related to the topic of the research work.
Author Response
Comments 1: The background is extensive and sufficient to understand the purpose of the research work. |
Response 1: Thank you very much
|
Comments 2: The topic of yellow silk cocoons is widely known, and the references contain various aspects of scientific interest, including the potential toxicological effects of sericin and its oligopeptide derivatives. Therefore, this work is interesting and relevant in its contribution. |
Response 2: Thank you very much
Comments 3: The toxicological evaluation places it on a level of scientific relevance, which increases interest and novelty in the research area, which is why I consider the topic relevant and important. Response 3: Thank you very much
Comments 4: The research methodology is clearly described, following conventional protocols, and it is relevant to highlight the use of international regulations for toxicological studies, which to date seek to minimize the use and sacrifice of animals. Response 4: Thank you very much
Comments 5: The bioethics committee authorization number for the use of animals is presented. Response 5: Thank you very much
Comments 6: With reference to the results, the biological evaluation tables are clear and descriptive. Response 6: Thank you very much
Comments 7: The graphs presented are legible and clearly show the description of the results in the text. Response 7: Thank you very much
Comments 8: The discussion directly supports each result obtained, which is why the conclusions are considered to be in accordance with each experiment or evaluation carried out. Response 8: Thank you very much
Comments 9: The writing of the work apparently does not require revision, since I felt it was fluent, without being an expert in languages. Response 9: Thank you very much
Comments 10: The references comply with the format requested by the journal, and I found no signs of plagiarism. Response 10: Thank you very much
Comments 11: I found several self-citations, about 13, however, they are directly related to the topic of the research work. Response 11: Thank you very much
Comments 12: I consider that the work can be published, without major revisions, since it is within the thematic framework of this journal. Response 12: Thank you very much
|

Reviewer 2 Report
Comments and Suggestions for Authors
Is the hypotensive effect considered a toxic effect? This should be indicated in the abstract section.
Introduction. Have there been any previous studies on Sericin-Derived Oligopeptides? This should stated in this section.
Do the authors have representative images of the histological analysis?
Was the protocol accepted by an ethical committee in 2006? When was this study carried out?
Was it safe for rodents to dissolve the Sericin-Derived Oligopeptides in distilled water and administrate this solution for weeks?
The microorganism's name should be in script Font (lines 137-139).
Was the ACCUCHEK system calibrated? How did the authors calibrate this equipment?
How was the pulse of the rat tail artery measured?
The term concentration should be changed by dose because the authors performed an in vivo study.
Lines 405-406. A previous study indicated that sericin can increase HDL levels. What is the molecular mechanism by which sericin increases the HDL levels?
Author Response
Comments 1: Is the hypotensive effect considered a toxic effect? This should be indicated in the abstract section. |
Response 1: For clarification, we have rewritten the abstract.
Line 12 – 25: “Sericin-derived oligopeptides (SDOs) from yellow silk cocoons exhibit antihypertensive and hypoglycemic properties in both in vitro and in vivo studies. This study investigated the acute toxicity of SDOs as a novel food for human consumption using female ICR mice and Wistar rats, as well as the chronic toxicity test on both sexes of Wistar rats. Clinical chemistry, hematology, and histopathological studies revealed that SDOs were safe for a single dose of 2000 body weight (BW) and daily oral administration of 50, 100, and 200 BW for six months. The chronic toxicity study additionally measured the rats' systolic blood pressure (SBP) and blood sugar monthly as they slowly aged. In the 2nd month for male rats and the 4th month for both sexes, SDOs had a significant hypotensive effect on Wistar rats' blood pressure, lowering it from 130 mmHg to a plateau at 110–115 mmHg. In contrast, the blood pressure of the control rats exceeded 140 mmHg after five months. Nonetheless, the hypoglycemic effect was not observed. Measurements of SBP and blood glucose in aged rats during chronic toxicity tests yielded insights beyond ordinary toxicity, including the health and fitness of the lab rats, perhaps resulting in novel discoveries or areas of study that justify the sacrifice of the animals' lives.”
|
Comments 2: Introduction. Have there been any previous studies on Sericin-Derived Oligopeptides? This should stated in this section. |
Response 2: Lines 35-37 present some previous research on SDOs from yellow silk cocoons. “Researchers discovered that sericin, sericin hydrolysates, and SDOs from yellow silk cocoons had hypoglycemic, antihypertensive, antioxidant, cholesterol-lowering, and colon cancer-reducing properties [5-12]”
Lines 51 and 52 provide detailed information on the effects of SDOs derived from yellow silk cocoons on the activity of ACE and DPP-IV inhibitors. “SDOs were found to inhibit the activity of ACE and/or DPP-IV in an in vitro model [7,27].”
Lines 62-64 include information on possible mechanisms of SDOs sourced from yellow silk cocoons, in addition to their ACE inhibitory action related to blood pressure reduction. “In an ex vivo study, SDOs from yellow silk cocoons demonstrated blood pressure lowering ability via NO/sGC/cGMP signal pathways in endothelium and the reduction of calcium influx in smooth muscle [6]”
Comments 3: Do the authors have representative images of the histological analysis? Response 3: Regrettably, we do not have them. In this case, we relied on the competence of a clinical pathologist, who would provide back the image that verified his examination result along with the report if he found the tissue to be abnormal. Generally, if the specimen is normal, we get just the examination report. In this acute toxicity experiment, he was tasked with examining 384 animal tissue slides positioned above his patients' specimens.
Following reviewer 3's suggestion, we have removed table 4 and explained the result in text form instead, and also merged the “Results” with “Discussion” to better present our findings.
Line 352-354 explained the histopathological results. “The histopathology of internal organs from the animals, including the heart, lung, liver, kidney, etc., did not reveal any lesions in either the control or the SDOs-treated groups, as reported by the clinical pathologist's examination.”
Comments 4: Was the protocol accepted by an ethical committee in 2006? When was this study carried out? Response 4: It was approved in 2012. The research project, undertaken from 2012 to 2017, examined the functional properties of yellow silk components, including silk lutein, sericin, and sericin-derived oligopeptides (SDOs), across multiple domains such as processing, toxicology, hypertension, diabetes, memory loss protection, immunomodulation, retinal cell protection, protection of primary human skin cells from UVB irradiation, formulation, and the epidemiology of age-related macular diseases (AMD) in Thailand. The results of this study have been prioritized for publication, since some studies may independently demonstrate the activity of SDOs, requiring corroborative evidence from further research. Several articles are still under preparation.
Comments 5: Was it safe for rodents to dissolve the Sericin-Derived Oligopeptides in distilled water and administrate this solution for weeks? Response 5: We rewrote lines 144–146 to make clear how we prepared the SDOs solution for animal treatment.
“We made SDOs solutions daily by mixing SDOs powder with distilled water to provide the required SDOs concentrations for oral administration (oral gavage) to the animals.”
Comments 6: The microorganism's name should be in script Font (lines 137-139). Response 6: Line 137 – 140: “Microbiological contamination includes Coliforms and E. coli (FDA BAM Online, 2017), Listeria monocytogenes (ISO 11290-1:2017 (E), Salmonella spp. (ISO 6579-1:2017 (E), Staphylococcus aureus (FDA BAM Online, 2016 (Chapter 12), Total Plate Count (FDA BAM Online, 2001 (Chapter 3), and Yeasts and Molds (FDA BAM Online 2001, Chapter 18).”
Comments 7: Was the ACCUCHEK system calibrated? How did the authors calibrate this equipment? Response 7: We did not perform any calibration on the Accu-Check test strips but relied on the manufacturer’s specification and protocol, similar to a study by Zordan et al. (2020), who also used Accu-Chek strips without further calibration. The test strips we used were within their valid date. In addition, a previous study by Meex et al. (2006) showed that the Accu-Check test strips for plasma typically gave results 7% higher than those for whole blood strips. To validate the strips' performance, 60 samples of venous whole blood were collected in EDTA tubes and compared with a blood gas analyzer. The study confirmed that plasma strips performed similarly to the glucose electrode of a blood gas analyzer, despite yielding slightly higher results. The Accu-Chek test strips, when used according to the manufacturer's guidelines, should provide reliable and consistent data for analysis, especially when compared to the blood gas analyzer used in previous research.
During the study, we randomly selected animals from different groups for sugar blood testing; if we found a statistically significant difference between each group, it would suggest a difference.
We rewrote lines 181–184 to specify the glucose test strips (Accu-Chek instant). “Blood glucose levels were then measured using glucose test strips (Accu-Chek Instant, Roche Laboratories, Pharma, Mannheim, Germany [53]) in accordance with the manufacturer's protocol and within the indicated validity period.”
Comments 8: How was the pulse of the rat tail artery measured? Response 8: According to the LE5001 non-invasive blood pressure meter, the pulse serves as a signal or indicator to initiate the measurement process. The rat's pulse needs to be at a sufficient level. If there is no signal or the signal level is lower than the necessary level (insufficient level), or if there is too much signal (pulse level high), the start button will not act. If the pulse level is correct (PULSE LEVEL READY), the start button will begin taking measurements.
Lines 188-191 provide further details on the pulse reading as a prerequisite for blood pressure assessment. “This was done to measure the pulse of the tail artery as a signal to initiate measurement. A sufficient level of pulse is required for the start button to function. If there is no signal or the pulse level is too high, the start button will not work.”
Comments 9: The term concentration should be changed by dose because the authors performed an in vivo study. Response 9: We have corrected it based on your suggestions.
Comments 10: Lines 405-406. A previous study indicated that sericin can increase HDL levels. What is the molecular mechanism by which sericin increases the HDL levels? Response 10: We have added a discussion from lines 300–318 on the molecular mechanism of how sericin increases HDL levels, as well as a comparison with SDOs.
“Results from a previous study indicated that sericin increased HDL levels in hypercholesterolemic rats via mitochondria function by enhancing its structure in the heart and liver [82]. Under normal conditions, HDL efficiently reverses cholesterol transport from peripheral tissues[78]. HDL's antioxidant activity in mitochondria protects LDL from oxidation, maintaining cardiovascular health. LDL controls cholesterol transfer to cells, reducing artery buildup [83]. HDL production and lipid metabolism depend on liver mitochondria, while cardiac mitochondria generate energy and optimize lipid utilization [84]. Since mitochondria generate reactive oxygen species (ROS), which damage them, HDL antioxidant activity is linked to mitochondrial health [85]. High oxidative stress may impair mitochondria [86], reducing the cell's antioxidant defenses and result in increased production of ROS. This deficit may lower the cell's antioxidant defenses and increase ROS generation. HDL and LDL can be oxidized by ROS at high amounts. Oxidized LDL is more likely to build up in artery walls, promoting plaque development and atherosclerosis, while oxidized HDL has lost its function on reverse cholesterol transfer and cardiovascular health [87,88]. SDOs may decrease the ROS level via their intrinsic antioxidant properties, similar to sericin but with higher efficacy, as Wu et al. (2008) shown that enzymatically hydrolyzed bioactive peptides of lower molecular weight from sericin exhibit superior antioxidant activities compared to intact sericin [89].”
New references added:
83. Barter, P., The role of HDL-cholesterol in preventing atherosclerotic disease. European Heart Journal Supplements 2005, 7 (suppl_F), F4-F8. 84. Han, Y. H.; Onufer, E. J.; Huang, L. H.; Sprung, R. W.; Davidson, W. S.; Czepielewski, R. S.; Wohltmann, M.; Sorci-Thomas, M. G.; Warner, B. W.; Randolph, G. J., Enterically derived high-density lipoprotein restrains liver injury through the portal vein. Science 2021, 373 (6553). 85. Murphy, M. P., How mitochondria produce reactive oxygen species. Biochem J 2009, 417 (1), 1-13. 86. Guo, C.; Sun, L.; Chen, X.; Zhang, D., Oxidative stress, mitochondrial damage and neurodegenerative diseases. Neural Regen Res 2013, 8 (21), 2003-14. 87. Kotani, K.; Sakane, N.; Ueda, M.; Mashiba, S.; Hayase, Y.; Tsuzaki, K.; Yamada, T.; Remaley, A. T., Oxidized high-density lipoprotein is associated with increased plasma glucose in non-diabetic dyslipidemic subjects. Clin Chim Acta 2012, 414, 125-129. 88. Heinecke, J. W., Lipoprotein oxidation in cardiovascular disease: chief culprit or innocent bystander? J Exp Med 2006, 203 (4), 813-6. 89. Wu, J.-H.; Wang, Z.; Xu, S.-Y., Enzymatic production of bioactive peptides from sericin recovered from silk industry wastewater. Process Biochemistry 2008, 43 (5), 480-487.
|

Reviewer 3 Report
Comments and Suggestions for Authors
Dear authors,
The manuscript entitled "Oral Toxicity and Hypotensive Influence of Sericin-Derived Oligopeptides (SDOs) from Yellow Silk Cocoons of Bombyx mori in Rodents Studies" investigated the acute toxicity of SDOs using female ICR mice and Wistar rats, anad chronic toxicity test on both sexes of Wistar rats. It presents scientific relevance for Medicine, Food, Pharmacy, Chemistry and others area. However, you need to change some details/information in the Abstract, Introduction, Material and Methods, results and discussion, and conclusions.
1. Abstract: Adequate! But:
- The abstract is well written, with information of the methods used. I suggest inserting more relevant results obtained (numerical data!!!).
- At the beginning, insert the meaning of the acronym "SDOs".
Please, to replace “mg/kg” by “mg kg-1” throughout the document, including figures and tables.
- At the end, I suggest highlighting the advantages/ disadvantages of the study and methods.
2. Introduction section:
- Adequate! I suggest inserting the reference:
- Please, to highlight the "innovative" proposal of the methods, as well as the advantages/disadvantages/limitations of the study.
3. Materials and methods section: The methodological proposal is appropriate to the manuscript, but I suggest:
- Page 3, in “2.1. Sericin-derived oligopeptides (SDOs) preparation” section: Long paragraph! I suggest dividing the text into 2 or 3 paragraphs.
- Page 4, in “Table 1”: the authors wrote “Heavy Metals”. It is not appropriate. The term “heavy metals” is the subject of many discussions. I suggest that the term be replaced, throughout the manuscript, according to [Science of the Total Environment 610–611 (2018) 419–420: “Heavy metal” - What to do now: To use or not to use?] and [Hazrat Ali & Ezzat Khan (2018) What are heavy metals? Long-standing controversy over the scientific use of the term ‘heavy metals’ – proposal of a comprehensive definition, Toxicological & Environmental Chemistry, 100:1, 6-19, DOI: 10.1080/02772248.2017.1413652]. Therefore, I strongly suggest removal of “heavy metals” from all text and replacement in the abstract and full text of submitted paper with words like “potentially toxic metal(s)/element(s)” or “trace metal(s)/element(s)”, according to the context, throughout the manuscript.
4. Results section (or “Results and discussion”)
Wouldn't it be more interesting to combine the "results” with the "discussion" to better describe the findings and compare them with other works published in the literature?? I suggest expanding the discussions!
- Page 8, in “Table 5”: I consider this table unnecessary! All values ​​are equal (0)! I suggest describing these results in text form!
- The results are interesting! I suggest discussing of the results obtained by comparing them with the literature!
5. Discussion section (or “Results and discussion”)?
Wouldn't it be more interesting to combine the "results” with the "discussion" to better describe the findings and compare them with other studies published in the literature?? I suggest expanding the discussions!
- If the authors choose not to combine the results with the discussions, I suggest dividing them into subsections (the same as the results) and discussing each subsection! I suggest expanding the discussions!
- I suggest, at the end of the "results and discussion", to write a paragraph summarizing the findings and their impacts on the research proposal.
6. Conclusion: I suggest inserting the results (numerical data!) obtained more relevant. I suggest pointing out the main results and disadvantages/limitations of the method and the study!
7. Table and Figures: Adequate. I suggest improving the concentration units of all figures!
8. References: Please, check if the references are in accordance with the journal's rules.
Comments on the Quality of English LanguageThe language (English) is satisfactory (but, I suggest the final revision)!
Author Response
Comments 1: The manuscript entitled "Oral Toxicity and Hypotensive Influence of Sericin-Derived Oligopeptides (SDOs) from Yellow Silk Cocoons of Bombyx mori in Rodents Studies" investigated the acute toxicity of SDOs using female ICR mice and Wistar rats, and chronic toxicity test on both sexes of Wistar rats. It presents scientific relevance for Medicine, Food, Pharmacy, Chemistry and others area. However, you need to change some details/information in the Abstract, Introduction, Material and Methods, results and discussion, and conclusions.
Abstract: Adequate! But: - The abstract is well written, with information of the methods used. I suggest inserting more relevant results obtained (numerical data!!!). - At the beginning, insert the meaning of the acronym "SDOs". Please, to replace “mg/kg” by “mg kg-1” throughout the document, including figures and tables. - At the end, I suggest highlighting the advantages/ disadvantages of the study and methods. |
Response 1: The abstract has been rewritten following your suggestions. Line 12 - 25: “Sericin-derived oligopeptides (SDOs) from yellow silk cocoons exhibit antihypertensive and hypoglycemic properties in both in vitro and in vivo studies. This study investigated the acute toxicity of SDOs as a novel food for human consumption using female ICR mice and Wistar rats, as well as the chronic toxicity test on both sexes of Wistar rats. Clinical chemistry, hematology, and histopathological studies revealed that SDOs were safe for a single dose of 2000 body weight (BW) and daily oral administration of 50, 100, and 200 BW for six months. The chronic toxicity study additionally measured the rats' systolic blood pressure (SBP) and blood sugar monthly as they slowly aged. In the 2nd month for male rats and the 4th month for both sexes, SDOs had a significant hypotensive effect on Wistar rats' blood pressure, lowering it from 130 mmHg to a plateau at 110–115 mmHg. In contrast, the blood pressure of the control rats exceeded 140 mmHg after five months. Nonetheless, the hypoglycemic effect was not observed. Measurements of SBP and blood glucose in aged rats during chronic toxicity tests yielded insights beyond ordinary toxicity, including the health and fitness of the lab rats, perhaps resulting in novel discoveries or areas of study that justify the sacrifice of the animals' lives.”
We have included the definition of the acronym "SDOs" in the abstract.
The unit “mg/kg” has been replaced with “mg kg-1” throughout the document, including figures and tables.
In the abstract, we highlighted in lines 22–25 the advantages of the study and method. “Measurements of SBP and blood glucose in aged rats during chronic toxicity tests yielded insights beyond ordinary toxicity, including the health and fitness of the lab rats, perhaps resulting in novel discoveries or areas of study that justify the sacrifice of the animals' lives.”
|
Comments 2: Introduction section: - Adequate! I suggest inserting the reference: - Please, to highlight the "innovative" proposal of the methods, as well as the advantages/disadvantages/limitations of the study. |
Response 2: We highlighted in the sentence in line 110 – 114.
“This study examined acute toxicity in female ICR mice and Wistar rats, alongside chronic toxicity in both sexes of Wistar rats. Additionally, the impact of prolonged SDOs consumption on the rats' blood glucose and blood pressure levels was assessed throughout the chronic toxicity testing period to evaluate the cumulative effects of SDOs as the rats aged gradually.”
Comments 3: Materials and methods section: The methodological proposal is appropriate to the manuscript, but I suggest: - Page 3, in “2.1. Sericin-derived oligopeptides (SDOs) preparation” section: Long paragraph! I suggest dividing the text into 2 or 3 paragraphs. - Page 4, in “Table 1”: the authors wrote “Heavy Metals”. It is not appropriate. The term “heavy metals” is the subject of many discussions. I suggest that the term be replaced, throughout the manuscript, according to [Science of the Total Environment 610–611 (2018) 419–420: “Heavy metal” - What to do now: To use or not to use?] and [Hazrat Ali & Ezzat Khan (2018) What are heavy metals? Long-standing controversy over the scientific use of the term ‘heavy metals’ – proposal of a comprehensive definition, Toxicological & Environmental Chemistry, 100:1, 6-19, DOI: 10.1080/02772248.2017.1413652]. Therefore, I strongly suggest removal of “heavy metals” from all text and replacement in the abstract and full text of submitted paper with words like “potentially toxic metal(s)/element(s)” or “trace metal(s)/element(s)”, according to the context, throughout the manuscript. Response 3: On page 3, we have divided the text into two paragraphs at lines 133-134.
Following your suggestion, we have replaced the word “heavy metals” with “trace metals" throughout the manuscript.
Comments 4: Results section (or “Results and discussion”) Wouldn't it be more interesting to combine the "results” with the "discussion" to better describe the findings and compare them with other works published in the literature?? I suggest expanding the discussions! - Page 8, in “Table 5”: I consider this table unnecessary! All values ​​are equal (0)! I suggest describing these results in text form! - The results are interesting! I suggest discussing of the results obtained by comparing them with the literature! Response 4: Based on your suggestion, we have merged the "results" with the "discussion".
We removed "Table 5" and described the results in text form. Line 352-354: “The histopathology of internal organs from the animals, including the heart, lung, liver, kidney, etc., did not reveal any lesions in either the control or the SDOs-treated groups, as reported by the clinical pathologist's examination.”
Comments 5: Discussion section (or “Results and discussion”)? Wouldn't it be more interesting to combine the "results” with the "discussion" to better describe the findings and compare them with other studies published in the literature?? I suggest expanding the discussions! - If the authors choose not to combine the results with the discussions, I suggest dividing them into subsections (the same as the results) and discussing each subsection! I suggest expanding the discussions! - I suggest, at the end of the "results and discussion", to write a paragraph summarizing the findings and their impacts on the research proposal. Response 5: We have written a new paragraph in line 598 – 617, summarizing the findings and their impacts on the study proposal.
“According to the findings of this study, SDOs exhibited either no acute toxicity at a fixed dose of 2000 BW or chronic toxicity at 50, 100, and 200 BW in both male and female Wistar rats. Through the integration of regular assessments of vital signs, such as blood pressure and blood sugar, during the 6-month chronic toxicity study, we were able to offer more comprehensive understanding into the health and well-being of the rats, both throughout their lifespan and as they aged. The duration of chronic toxicity often extends up to six months, representing around 25% of the typical lifespan of the experimental rats. The data obtained from live and aging rats, especially in terms of SBP and blood glucose levels, may provide insights into the influence of oral administration of SDOs at low dosages and indicate that SDOs could potentially contribute to general health and fitness related to longevity. Previous investigations on SDOs have demonstrated a variety of bifunctionalities and mechanisms of action over the past few decades. Based on this information, we propose that the inherent strong antioxidant activity of SDOs, their inhibitory effects on pancreatic and ACE enzymes, and their enzymatic resistance in the gastrointestinal tract and bloodstream provide bene-fits for their transport into the circulatory system and their functional expression at various receptive sites in multiple internal organs, including mitochondria, thereby opening up novel research approaches. Measuring vital signs in live and aging laboratory animals could be labor-intensive, time-consuming, and necessitates appropriate facilities, especially when evaluating a substantial cohort of animals.”
Comments 6: Conclusion: I suggest inserting the results (numerical data!) obtained more relevant. I suggest pointing out the main results and disadvantages/limitations of the method and the study! Response 6: The conclusion has been rewritten. Line 629 – 644: “Rats and mice, the two experimental female animal species tested for acute toxicity, did not exhibit signs or symptoms of SDOs from yellow silk cocoons. Histopathological examination of the internal organs revealed no abnormalities in any of the animals. The research also found no significant effects on clinical chemistry, with the exception of uric acid, cholesterol, HDL, albumin, and globulin levels. This could potentially be due to the high dosages of SDOs, which could potentially damage liver function. Male and female rats showed no abnormalities throughout the chronic toxicity test. Moreover, all SDOs-treated rats had lowered mean SBP from 130 mmHg and remained in the young healthy range of 110–115 mmHg after four months, while the control rats continued to have increased SBP as they aged and reached over 140 mmHg at the end of the study period, suggesting hypertensive onset. The SDOs-treated rats had slightly lower blood sugar levels than the control, but there were no significant differences. The chronic toxicity, along with periodic checks of SBP and blood sugar levels in live rats as they aged, provided useful information on the hypotensive effects of SDOs on aging rats. This led to an investigation on hypertensive rats, but it could be time-consuming, labor-demanding, and require specific measurement equipment and facilities.”
Comments 7: Table and Figures: Adequate. I suggest improving the concentration units of all figures! Response 7: We have improved the concentrations and doses units following your suggestions.
Comments 8: References: Please, check if the references are in accordance with the journal's rules. Response 8: We have corrected per your suggestions in accordance with the journal’s rules.
|
4. Response to Comments on the Quality of English Language |
Point 1: The language (English) is satisfactory (but, I suggest the final revision)! |
Response 1: We have revised the manuscript per your suggestion.
|

Round 2
Reviewer 2 Report
Comments and Suggestions for Authors
The manuscript can be accepted for publication